# Efficacy and Safety of First-Line Targeted Treatment and Immunotherapy for Patients with Biliary Tract Cancer: A Systematic Review and Meta-Analysis

**DOI:** 10.3390/cancers15010039

**Published:** 2022-12-21

**Authors:** Xin Yan, Huimin Zou, Yunfeng Lai, Carolina Oi Lam Ung, Hao Hu

**Affiliations:** 1State Key Laboratory of Quality Research in Chinese Medicine, Institute of Chinese Medical Sciences, University of Macau, Macao SAR, China; 2School of Public Health and Management, Guangzhou University of Chinese Medicine, Guangzhou 510006, China; 3Department of Public Health and Medicinal Administration, Faculty of Health Sciences, University of Macau, Macao SAR, China

**Keywords:** biliary tract cancer, cholangiocarcinoma, targeted therapy, immunotherapy

## Abstract

**Simple Summary:**

We performed a meta-analysis of all clinical trials of first-line combination therapies for advanced biliary tract cancer and concluded that combining immunotherapy with chemotherapies could improve survival for patients with aBTCs by increasing the objective response rate.

**Abstract:**

Background: Biliary tract cancer is one of the most aggressive and fatal tumours. Gemcitabine with cisplatin chemotherapy has long been the first-line treatment, but the prognosis is poor. In recent years, targeted treatment and immunotherapy have produced encouraging outcomes requiring a thorough review and meta-analysis. Method: For this systematic review and meta-analysis, we searched four databases, starting from the inception dates of databases to 11 January 2022. This study comprised randomised clinical trials and cohort studies that used immunotherapy or targeted treatment as the first line of treatment for patients with biliary tract cancer. Results: From the 888 studies extracted, 33 trials were examined and found to meet the criteria. These included 3087 patients, 16 single-arm trials, 13 RCTs, one nRCT, a prospective single-arm pilot study, and a clinical setting in the real world. From 2010 to 2020, 33 studies were conducted using targeted treatment or immunologic therapies as first-line treatments for BTC patients, and 18 of those studies had positive outcomes. Conclusion: This study demonstrates that immunotherapy combined with chemotherapy as first-line treatment can provide survival benefits by improving the objective response rate for patients with unresectable biliary tract cancer. The potential for combination therapy to become a new trend in clinical treatment is promising but needs further clinical evaluation.

## 1. Introduction

Biliary tract cancer (BTC), which includes intrahepatic cholangiocarcinoma (ICC) and extrahepatic cholangiocarcinoma (ECC), is one of the most malignant and lethal tumours (ECC). The epidemiological characteristics of ICC and ECC differ. ECC is the most common subtype of cholangiocarcinoma in general. However, ICC is more common in some East Asian countries, accounting for 15% to 20% of all liver cancers and typically invading the bile duct wall [1]. In Western countries, the incidence and mortality rate of ICC are increasing [2]. There are significant gender and ethnic differences in ICC incidence and mortality. Men are 1.5 times more likely than women to develop ICC. Asians have a 2.0 times higher incidence rate than whites and blacks. Southeast Asia and China have the highest prevalence of ICC worldwide. Indian Americans, Alaska Natives, and Asians have the highest ICC mortality rates, while Caucasians and blacks have the lowest [3].

Most BTC patients cannot be resected at the time of diagnosis, and the prognosis is poor, with a median survival of 3–6 months, compared to 10% and 0% 5-year survival rates for stage III and IV BTC, respectively [2].

Currently, surgery is the only curable treatment option for cholangiocarcinoma that has not spread beyond the primary site. The objective response rate (ORR) is 15–26%, with a median survival time of less than one year, and drug resistance is common [1]. Most patients with ICC are initially diagnosed with local invasion or distant metastasis and do not have the option of undergoing radical surgery. In recent years, the morbidity and incidence of intrahepatic BTC have increased, while extrahepatic BTC has decreased. 

Cisplatin and gemcitabine are the standard first-line chemotherapy treatment for BTC [4]. Most BTC patients have no other treatment options after developing resistance to first-line chemotherapy, and their disease often worsens rapidly. The dismal median overall survival (median-OS) of 11–13 months under systemic palliative therapy with gemcitabine and cisplatin highlights the urgent need to expand the limited therapeutic measures available to date for patients with advanced BTC [5].

Patients have benefited from the emergence of targeted therapeutic agents in recent years. Mutations in FGFR, IDH, BRAF, and NTRK are all linked to the development of BTC. Fibroblast growth factor receptor (FGFR) and neurotrophic tyrosine receptor kinase (NTRK) have been linked to cholangiocarcinoma development and are expected to be important targeted therapies [6,7]. Recent sequencing results from multiple sources have shown that up to 11–45% of patients with ICC contain FGFR2 fusion mutations. The binding proteins of fusion mutations include ARID1A-, PBRM1-, and TP53-. In addition, 24 patients with FGFR2 fusion mutations were reported in the MSKCC 10,000 sequencing data, including 18 cases of cholangiocarcinoma (242 cases in total), accounting for 75% of all FGFR2 fusion mutations. It is evident that FGFR2 fusion mutations are relatively highly enriched in cholangiocarcinoma. To summarise previous reports, IDH mutations in cholangiocarcinoma have the following characteristics: (1) IDH1 mutations are more frequent than IDH2 mutations; the hotspot mutation of IDH1 is located at R132, while the hotspot mutation of IDH2 is located at R172; (2) the proportion of mutations is higher in ICC than in ECC; and (3) IDH1/2 mutations lose normal enzymatic activity and generate new activity, which can produce the oncogenic metabolite 2-hydroxyglutarate (2-HG). 2-HG can be detected in tumours or blood and can be used as one of the PD indicators in clinical trials [7]. Other targets, such as anti-angiogenesis, EGFR amp, WNT/a-catenin, Hedgehog, and HGF/c-MET, have been reported in cholangiocarcinoma, but most of these pathways can be found in most tumour types, and multiple previous clinical trials in cholangiocarcinoma have shown limited effectiveness.

ICIs monotherapy has achieved some efficacy in patients with cholangiocarcinoma. On this basis, several clinical trials are underway investigating the combination of ICIs or ICIs with other types of immunotherapies. CTLA-4 regulates early immune responses, and PD-L1 mainly regulates immune responses in advanced peripheral tissues. Based on this regulatory mechanism, the combination of immunosuppressive agents of PD-1, PD-L1, and CTLA-4 can be made to achieve antitumour therapy through complementary mechanisms with synergistic effects. It has been shown that the combination of CTLA-4 and PD-1 inhibitors is more effective than single therapy, probably because the synergistic effect leads to an increase in the number of tumour-infiltrating lymphocytes, a decrease in regulatory T cells, and an overall improvement in the inhibition of tumour growth, with an overall efficiency of 10.8% (7/65, all in partial remission), a disease control rate of 32.2%, and an overall survival time for cholangiocarcinoma was 10.1 months [8].

Furthermore, while there is no evidence of adjuvant therapy, many studies on treating BTC with immune checkpoint inhibitors have been conducted (ICI). For the first-line treatment of advanced BTC, a combination of ICI and chemotherapy or targeted therapies are still being studied in clinical trials [9].

However, there is no comprehensive systematic review or meta-analysis of the efficacy of updated BTC treatment. Only EGFR inhibitors were studied in Alexandro Rizzo’s study [10]. Although data on treatment, including systemic chemotherapy and radiation therapy, were analysed in Michael N’s study, the data were only available up to 2013 and did not focus on targeted therapies [11]. As a result, there is a need for more recent and comprehensive meta-analysis in this field.

Through systematic review and meta-analysis, this study aimed to summarise the evidence from randomised controlled trials (RCTs) and cohort studies comparing the efficacy and safety of immunotherapy and targeted therapy for BTCs. The findings of this study are expected to provide evidence for managing patients with BTCs.

## 2. Methods

### 2.1. Study Design 

This systematic review and meta-analysis were conducted and reported under the Preferred Reporting Items for System Review and Meta-Analysis Scenarios (PRISMA) [12]. It was registered with PROSPERO (CRD42022336576).

Extensive searches of databases for clinical trials related to BTC were conducted. The inclusion/exclusion criteria for this study were developed in accordance with the PICOS Principles [13].

### 2.2. Inclusion Criteria for Study Selection

#### 2.2.1. Types of Studies

We focused on RCT and cohort studies. Case-control studies, letters, reviews, case reports, and articles that do not provide raw data were not included. When data came from different phases of the same experiment, only studies with the most complete and up-to-date data were retained. No restrictions were placed on the language of the article. We also searched CNKI but did not find any more additional Chinese studies. 

#### 2.2.2. Population

This study targeted populations who needed to meet all the following criteria:

(1) Unresectable gallbladder cancer and cholangiocarcinoma diagnosed by histopathology or cytology.

(2) The patient has not received systemic treatment for unresectable biliary cancer.

(3) The patient has at least one measurable lesion.

#### 2.2.3. Interventions

Targeted and immunologic agents used alone or in combination with chemotherapeutic agents were included in this study. The drug targets include MEK1/2, EGFR, VEGF, mTOR, PD-1/PD-L1, MET, and CTLA-4 (as shown in Table 1).

#### 2.2.4. Outcome Measures

##### Primary Outcomes

The primary outcomes were the objective response rate (ORR), disease control rate (DCR), overall survival (OS), and progression-free survival (PFS). 

Overall Survival (OS) is the time between randomisation and the onset of death by any cause. Objective Response Rate (ORR) is the proportion of patients whose tumour volume decreases to a predetermined value. ORR equals the ratio of complete response (CR) to partial response (PR), or ORR = CR + PR. ORR excludes both stable disease (SD) and the effect of the disease’s natural progression. Smaller sample sizes and shorter follow-up periods are required. Partial response (PR) is defined as a volume reduction of at least 30 percent in all tumours that can be measured. The Disease Control Rate (DCR) is the proportion of patients whose cancer diminishes or stabilises over time. DCR equals the sum of the rates of complete remission, partial remission, and stable disease. PFS is the time between randomisation and the onset of objective tumour progression or death from any cause, which is a surrogate endpoint for OS.

##### Secondary Outcomes

Secondary outcomes were treatment-related adverse events (TRAE). 

### 2.3. Data Sources and Search Strategy 

Two investigators searched PubMed, Web of Science, Embase, Cochrane Library, and clinicaltrials.gov, starting from the inception dates of databases to 11 September 2022. The terms “biliary tract cancer”, “cholangiocarcinoma”, gene mutation type such as “EGFR”, and drug names such as “pembrolizumab” were used as keywords to search titles or abstracts (See Appendix A for a detailed search strategy).

### 2.4. Literature Selection

Endnote 20 software was used to import all the search results. First, duplicates were removed. Second, two researchers independently screened titles and abstracts to determine inclusion eligibility. Third, full manuscripts of potentially eligible trials were read to determine which studies should be included. During the literature selection process, disagreements were resolved through discussion between the two researchers with the assistance of a third researcher as needed [14].

### 2.5. Data Extraction

Two researchers extracted and compiled data. The first author, study method, publication time, journal of publication, follow-up time, number of patients, baseline level of patients, observed indicators, and interventions were all extracted from the study data. If there were any disagreements, the researchers discussed them. If two researchers cannot reach an agreement, a third researcher makes the final decision. We recorded the data in Microsoft Excel.

### 2.6. Quality Assessment 

The PRISMA guidelines were followed when conducting the systematic review. The quality of the included literature was assessed using Version 2 of the Cochrane risk-of-bias tool for randomised trials and plotted in the quality evaluation table [15]. Non-randomised intervention studies were evaluated using ROBINS-I (Risk Of Bias In Non-randomised Studies of Interventions) [16].

### 2.7. Statistical Analysis

We used the “meta” package in R environment to perform the meta-analysis using frequentist approach [17]. Heterogeneity was assessed with *I*^2^. A fixed-effects model was used for small heterogeneity (*I*^2^ < 25%), and a random-effects model was used for a large one (*I*^2^ > 25%) [18]. Bias is tested using funnel plots if the heterogeneity between studies in the meta was too large (*I*^2^ > 75%) [19]. We determined the rank of all interventions by using the ‘netrank’ function in the ‘netmeta’ package in R to obtain P-scores. The P-score ranged from 0 to 1, showing a progressive rise in the efficacy of the included medications based on the estimation result and confidence interval of the effect value [18].

## 3. Results

### 3.1. Study Inclusion

We initially identified a total of 888 articles from four databases. After excluding 179 duplicates, 709 articles remained. Then, we screened titles and abstracts to exclude the types of articles that did not meet the criteria, leaving 128 articles. Full-text screening was performed to exclude 73 articles with immunotherapies or targeted therapies as second-line therapy, Seven articles with a mix of first-line and second-line therapy, and eight articles whose studies included other diseases were excluded, leaving 33 included. After searching clinicaltrial.gov and checking the reference of the included literature, 17 additional studies were identified, resulting in a final 33 studies that met all the inclusion criteria (Figure 1). The total number of patients included in the trials is 3087.

### 3.2. Characteristics of the Included Studies

#### 3.2.1. Targeted Therapies

This study included 22 trials focusing on targeted therapies, with a total of 1658 patients. The basic characteristics of patients are listed in Table 2. 

Among the 22 studies evaluating targeted therapy, 11 studies were single-arm phase II trials; 10 studies were randomised parallel phase II trials, and one study was a phase III trial. The publication years ranged from 2010 to 2021, with the earliest trial starting in 2006. There were 11 single-arm trials, 11 controlled trials, one with three groups (Valle et al., 2020 [20]), and three trials were blinded (Santoro et al., 2015 [21]; Valle et al., 2015 [22]; Moehler et al., 2014 [23]). One nRCT (Factorial assignment) and 10 RCTs were included. Seven trials were completed in the United States; five were multicenter trials, and the others were distributed in Italy, Australia, Austria, France, Korea, Taiwan, Denmark, and Germany.

**Table 2 cancers-15-00039-t002:** Basic characteristics of targeted therapies.

No.	Study	Phase	pts	Location	Intervention	Dose	Chemotherapy	Mean Age	Gender	Race
1	Khoueiry (2012) [24]	II	31	US	Sorafenib	400 mg po twice daily continuously.	NA	57.8 (33–81)	male: 15 (48%)	– White: 24 (77%) – Black: 4 (13%) – Multi-racial 1 (3%) – Native American 1 (3%) – Unknown 1 (3%)
2	Khoueiry (2014) [25]	II	34	US	Sorafenib	400 mg BID and 100 mg daily	NA	63	male: 13 (38%)	– White 28 (82%) – Black 3 (9%) – Asian 1 (3%) – Native American 1 (3%) – Unknown 2 (6%)
3	Hezel (2014) [26]	II	31	US	Panitumumab	6 mg/kg	GEMOX	NA	NA	NA
4	Santoro (2015) [21]	II	173	Italy	Vandetanib	Vandetanib (300 mg or 100 mg) or placebo was given in single oral daily doses.	Gemcitabine	63.6 (sd: 9.5)	male: 81 (46.8)	– White: 170 (98.3%)
5	Zhu (2010) [25]	II	35	US	Bevacizumab	10 m g/kg	GEMOX	NA	NA	NA
6	Gruenberger (2010) [27]	II	30	Austria	Cetuximab	500 mg/m^2^	GEMOX	median age: 68 years (IQR 62–73)	NA	NA
7	Lau (2018) [28]	II	27	Australia	Everolimus	10 mg/d	NA	NA	NA	NA
8	Malka (2014) [29]	II	150	France	Cetuximab		GEMOX	NA	NA	NA
9	Sohal (2013) [30]	II	35	US	Panitumumab	9 mg/kg	Gemcitabine Irinotecan	NA	NA	NA
10	Borbath (2013) [31]	II	44	Multi- center	Cetuximab	400 mg/m^2^ at week 1, then 250 mg/m^2^/week	GEM	median age: 61.5	NA	NA
11	Lee (2013) [32]	II	39	US	Sorafenib	400 mg twice daily	GEMCIS	NA	NA	NA
12	Lee (2012) [33]	III	−135 −133	Korea	Erlotinib		GEMOX	chemotherapy alone: 61 (55–68) C + T: 59 (54–66)	male: A: 79 (59%) B: 91 (67%)	NA
13	Chen (2015) [34]	II	−62 −60	Taiwan	Cetuximab	C-GEMOX (500 mg/m^2^ cetuximab plus GEMOX) every 2 weeks	GEMOX	C-GEMOX: 61 (32–78) GEMOX: 59 (32–80)	male: C-GEMOX: 28 (45%) GEMOX: 30 (50%)	NA
14	Valle (2015) [22]	II	62	Multi -center	Cediranib		GEMCIS	NA	NA	NA
15	Valle (2020) [20]	II	106 102 101	Multi -center	Ramucirumab		GEMCIS	NA	NA	NA
16	Jensen (2012) [35]	II	46	Denmark	Panitumumab		GEMOX	NA	NA	NA
17	Leone (2016) [36]	II	45 44	Italy	Panitumumab		GEMOX	NA	NA	NA
18	Lowery (2019) [37]	II	41	US	Binimetinib	45 mg orally twice daily	GEMCIS	66 (45–83)	male: 21 (51.2%)	NA
19	Moehler (2014) [23]	II	102	Germany	Sorafenib	400 mg bid orally continuously	GEM	Sorafenib: 64.0 placebo: 64.5	sorafenib: male: 20 Gemcitabine: male: 23	– European (100%)
20	Iyer (2018) [38]	II	50	Multi -center	Bevacizumab		Gemcitabine Capecitabine	NA	NA	NA
21	Lubner (2010) [39]	II	53	Multi -center	Bevacizumab Erlotinib		NA	63 (31–87)	male: 23 (43%)	NA
22	Vogel (2018) [40]	II	−62 −28	Germany	Panitumumab	9 mg/kg BW at day 1	GEMCIS	NA	NA	– White 91% – Asian 9%

The primary tumour sites are shown in Table 3 below. 694 patients with IHC, 384 patients with gallbladder cancer, 235 patients with EHC, 44 patients with hilar cholangiocarcinoma, and 17 patients with Vater ampulla carcinoma were included. Three studies (consisting of 219 patients) did not specify the disease type in patients with BTC, and 48 patients were classified as “other disease types” in the original literature, which included patients with liver metastases.

#### 3.2.2. Immunotherapies

As shown in Table 4, the included immunotherapy-related studies were published between 2018 and 2021. The trial of Oh et al., 2020 [41] of Bintrafusp alfa is a phase 2/3 trial. Oh et al., 2022 [42] is a phase 3 trial, demonstrating the effectiveness of Durvalumab in combination with chemotherapy therapies. No immunotherapy was used alone in any of the study’s regimens, which all combined both immunotherapy and chemotherapy as the first-line treatment. A total of 1286 patients were included. 

The primary tumour sites of immunotherapy in the studies are listed in Table 5, among which most were ICC, 460 cases, accounting for 50.9% of the total. 

### 3.3. Quality Assessment

Figure 2 and Figure 3 depict the quality of the included RCTs. Overall, the included randomised controlled trials were of high quality. El-Khoueiry (2012) [24] was terminated after the first phase of accrual because the primary objectives were not met. Iyer (2018) [38] is a meeting abstract lacking detailed data.

As shown in Table 6, ROBINS-1 assessed the quality of 16 single-arm studies and observational studies. All studies documented the definition of controls and the comparability of cases and controls, except for one study that did not report case definitions (Khoueiry, 2012 [24]).

### 3.4. Systematic Review

#### 3.4.1. Targeted Therapies

Zhu et al., 2010 [52] showed that patients with BTC treated with bevacizumab + GEMOX obtained a median PFS of 7.0 months (95% CI 5.3–10.3) and a 6-month PFS of 63% (47–79), below the set target rate of 70%, with objective responses recorded in 19 patients and an overall disease control rate of 80%. The trial by Gruenberger et al., 2010 [27] had an overall disease control rate of 80%, progression-free survival of 8.8 months (95% CI 5.1–12.5), and median overall survival of 15.2 months (9.9–20.5) for all treated patients. The trial by Lau et al., 2018 [28], using Everolimus 10 mg/d alone, had a median PFS of 5.5 months (95% confidence interval (CI: 2.1–10.0 months) and a median OS of 9.5 months (95% CI: 5.5–16.6 months). Notably, gallbladder cancer had a significantly worse DCR at 12 weeks than other anatomic sites and a trend toward worse PFS and OS, but the treatment was well tolerated.

Sohal et al., 2013 [30] added Irinotecan and Panitumumab to Gemcitabine, which had a median PFS of 9.7 months and a median OS of 12.9 months, showing encouraging efficacy and good tolerability of this regimen. The trial by Borbath et al., 2013 [31] met the primary endpoint with a median PFS time of 5.8 months (95% CI 3.6–8.5 months), median OS time of 13.5 months (95% CI 9.8–31.8 months), and 53.7% of patients remained alive at 1 year, suggesting that Gemcitabine-Cetuximab has activity in BTC and that KRAS status is not associated with PFS and, unlike cutaneous toxic effects, may serve as a surrogate marker of efficacy. Lubner et al., 2010 [39] showed that 87% of patients showed disease progression, with a median time to disease progression of 4.4 months and a median OS of 9.9 months. This study concluded that the combination of bevacizumab and erlotinib demonstrated significant activity in treating advanced BTC, with few adverse events of grades 3 or 4. Bevacizumab and Erlotinib demonstrated significant activity in advanced BTC with few Grade 3 or 4 adverse events. The trial by Leone et al., 2016 [36] added Panitumumab to GEMOX, and the results confirmed a marginal effect of anti-EGFR therapy in WT-KRAS-selected BTC.

Lee et al., 2012 [33] combined erlotinib with GEMOX and showed no significant difference in progression-free survival between the groups. Still, adding erlotinib to gemcitabine and oxaliplatin showed antitumour activity: significantly more patients had objective responses in the chemotherapy plus erlotinib group than in the chemotherapy alone group (40 patients versus 21 patients; *p* = 0.005), but median overall survival was the same in both groups. The trial by Chen et al., 2015 [34] showed a trend toward improved PFS was observed, but the addition of cetuximab did not significantly improve the ORR of GEMOX chemotherapy in advanced BTC, and KRAS mutations did not affect the trend in ORR and PFS differences between C-GEMOX and GEMOX. In the study by Jensen et al., 2012 [35], the addition of Panitumumab to chemotherapy resulted in a 6-month progression-free survival (PFS) rate of 31/42 [74%; 95% confidence interval (CI) 58% to 84%], a disease control rate of 86%, a median PFS of 8.3 months (95% CI 6.7–8.7 months), and a median overall survival of 10.0 months (95% CI. 7.4–12.7 months). Hezel et al., 2014 [26] used a combination of gemcitabine, oxaliplatin, and panitumumab for KRAS wild-type metastatic BTC and achieved a remission rate of 45% and a disease control rate of 90%. Its median PFS was 10.6 months (95% CI 5–24 months), and median overall survival was 20.3 months (95% CI 9–25 months).

Other trials did not meet the expected endpoints but were still informative. Khoueiry et al., 2012 [24], as a phase II study of sorafenib in patients with advanced BTC based on the role of the RAS-RAF-MEK-ERK pathway and the VEGF axis in BTC, was terminated after phase I due to failure to meet the primary objective. The trial by Khoueiry et al., 2014 [25] to study sorafenib and erlotinib was also terminated after Phase I enrollment. Lee (2012) [33] added sorafenib to gemcitabine and cisplatin for biliary tract adenocarcinoma, which did not improve efficacy compared with historical data and had increased toxicity. In the trial of Santoro et al., 2015 [21], patients were randomised in a 1:1:1. The results showed no statistical difference between secondary endpoints except for ORR, and the V/G combination was slightly outperformed by the other treatments. Patients in the three groups reported similar rates of adverse effects. Malka et al., 2014 [29] concluded that adding cetuximab to gemcitabine and oxaliplatin did not appear to enhance chemotherapeutic activity in patients with advanced BTC, although it was well tolerated. 

The trial by Valle et al., 2015 [22] showed that Cediranib did not improve progression-free survival in patients with advanced BTC in combination with cisplatin and gemcitabine. Valle et al., 2020 [20] added Ramucirumab or Merestinib to GEM + CIS standard chemotherapy and showed no improvement in PFS, OS, or ORR. Lowery et al., 2019 [37] demonstrated that Binimetinib, combined with Gemcitabine and Cisplatin, had no effect on PFS-6-month or RR. Moehler et al., 2014 [23] similarly demonstrated that adding Sorafenib to Gemcitabine did not improve outcomes in patients with advanced BTC, but biomarker subgroup analysis suggested that some patients may benefit from the combination. Iyer et al., 2018 [38] demonstrated that adding Bevacizumab to Gemcitabine/Capecitabine did not improve prognosis in unselected patients with advanced BTC compared to historical controls. Vogel et al., 2018 [40] concluded that combining Panitumumab with chemotherapy did not improve ORR, PFS, or OS in patients with KRAS wild-type advanced BTC.

#### 3.4.2. Immunotherapies

The trial of Yu et al., 2021 [43] resulted in an ORR of 14.3% (95% CI: 1.8 to 42.8), a DCR of 64.3% (95% CI: 41.7 to 86.9), a median PFS of 6.5 months (95% CI: 3.8 to 9.2), PFS rates of 61.6% and 12.3% at 6 and 12 months, respectively, and a median OS of 9.9 months (95% CI: 7.6 to 12.2), concluding that Camrelizumab in combination with chemotherapy as first-line treatment for metastatic BTC demonstrated acceptable safety and efficacy. Chen et al., 2015 [34] also concluded that Camrelizumab plus GEMOX as first-line treatment for patients with advanced BTC looked promising, with a median PFS that was 6.1 months and median OS that was 11.8 months. Sun et al., 2018 [44] concluded that the combination of PD-1 antagonist plus chemotherapy or targeted therapy was effective and tolerable as first-line treatment for advanced BTC. OS was significantly longer in the group treated with the combination drug than in the chemotherapy group (median, 8.2 vs. 3.6 months, HR 0.47 [0.20–1.10], *p* = 0.011), as was PFS (median, 3.9 vs. 2.0 months, HR 0.58 [0.28–1.19], *p* = 0.034), *p* = 0.034), and no significant ORR difference was observed.

Gou et al., 2021 [45] yielded results that in advanced BTC, anti-PD-1 therapy plus chemotherapy prolonged PFS compared to chemotherapy alone, and AE was tolerable. Oh et al., 2020 [46] showed that adding D + T immunotherapy to chemotherapy was tolerable and showed promising efficacy. Chiang et al., 2021 [48] concluded that Nivolumab in combination with a modified GS (gemcitabine and S-1) is a promising regimen with a good safety profile. Oh et al., 2020 [41] showed that Bintrafusp alfa was clinically active in Asian patients with BTC and had a durable response. Oh et al., 2022 [42] concluded that Durvalumab in combination with chemotherapy for BTC significantly improved OS compared to chemotherapy alone (D: 12.8 (11.1–14.0) vs. placebo: 11.5 (10.1–12.5)). The combination also greatly improved progression-free survival compared with chemotherapy alone. Median progression-free survival with durvalumab combined with gemcitabine and cisplatin was 7.2 months compared with 5.7 months with chemotherapy alone (HR = 0.75; *p* = 0.001). The proportion of progression-free patients was 34.8% and 24.6% at 9 months and 16.0% and 6.6% at 12 months, respectively. The ORR also improved, with an overall efficacy rate of 26.7% with durvalumab/chemotherapy compared with 18.7% with chemotherapy alone, with a superiority ratio of 1.60 for efficacy (*p* = 0.011). The trial by Sahai et al., 2020 [49] concluded that the combination of nivolumab with chemotherapy drugs failed to improve efficacy.

#### 3.4.3. Combined Therapies

In the study by J. Zhou et al. [53], 30 patients with advanced ICC were included with an ORR of 80% (24/30; 95% CI: 61.4–92.3%) and a DCR of 93.3% (28/30; 95% CI: 77.4–99.2%). A complete response (CR) was scored 1. The median duration of follow-up was 8.4 months. Twelve patients experienced disease progression, and four patients died. Median PFS and OS had not been reached. The median duration of response has not been determined, and the 6-month OS rate was 90%. A quantity of 43% (13/30) of patients experienced grade 3 or higher adverse events (AEs). This study showed that ORR was significantly associated with PD-L1 expression and mutations associated with DNA damage repair (DDR) in tumour samples. In patients with advanced ICC, the combination of toripalimab, lenvatinib, and GEMOX chemotherapy was well tolerated and showed an encouraging ORR.

In the prospective phase II trial by Q. Zhang et al. [54], the efficacy and safety of first-line lenvatinib plus PD-1 inhibitor was similarly evaluated in patients with initially unresectable BTC, and the feasibility of translational surgery after this treatment was explored. The study included 38 patients, with a mean age of 62.5 years and 14 men, receiving PD-1 inhibitors, including Pembrolizumab (7.9%), Toripalimab 12 (31.6%), Tislelizumab 11 (28.9%), Sintilimab 11 (28.9%), and Camrelizumab 1 (2.6%), after a median follow-up of 13.7 (95% CI: 9.7 to 17.8) months, the 1-year OS rate was 47.4% (18/38), and 65.8% of patients were still alive. Median EFS was 8.0 months (95% CI: 4.6 to 11.4), and median OS was 17.7 months (95% CI: not estimable). Among the 13 patients who underwent conversion surgery, the median EFS was 13.5 months (95% CI: 13.0 to 14.0). Among patients who received only systemic therapy, the median EFS was 4.6 months (95% CI: 0.8 to 8.4). and the median OS was 12.4 months (95% CI: 8.5 to 16.3).

### 3.5. Meta-Analysis for OS

#### 3.5.1. Meta-Analysis for OS of Targeted Therapy

As the heterogeneity was considerably high (*I*^2^ = 63%), a random effects model was used to obtain a pooled OS of 10.65 months for the targeted drug treatment group (Figure 4).

As evidenced by the funnel plot with a more symmetrical distribution, these studies have less publication bias. Egger’s test for a regression intercept gave a *p*-value of 0.1377 > 0.05, indicating no evidence of publication bias (Figure 5).

A subgroup meta-analysis of the chemotherapy and combined therapy with targeted therapy, yielded *I*^2^ = 65%, *p*-value = 0.21 (Figure 6). So, a difference between the two groups could not be demonstrated.

The asymmetry of the funnel plot indicates a possible publication bias (Figure 7). The Egger’s test could not be applied because the sample size was less than 10.

#### 3.5.2. Meta-Analysis for OS of Immunotherapy

The pooled overall survival is 15.62 months for the immunotherapy group, because the heterogeneity *I*^2^ = 84% > 50%, random effects model was applied (Figure 8). 

### 3.6. Meta-Analysis for PFS

#### 3.6.1. Meta-Analysis for PFS of Targeted Therapies

Data on PFS were available for six studies involving immunotherapy and 17 studies involving targeted treatment. Plots of each study’s individual PFS and its confidence intervals were made for the two groups (Figure 9). In the targeted therapy group, the median PFS was 6.02 months (95% CI: 5.01–7.03) (range: 2.0–13.5 months).

As evidenced by the funnel plot with a more symmetrical distribution, these studies have less publication bias (Figure 10). In addition, Egger’s test for a regression intercept gave a *p*-value of 0.3583 > 0.05, indicating no evidence of publication bias.

Further subgroup analysis was performed to divide the studies of targeted therapies into two groups, EGFR and VEGF, and the pooled PFS was obtained from the forest plot (Figure 11 and Figure 12). After Egger’s test, the *p*-value of EGFR was 0.9437 > 0.05, and the *p*-value of VEGF was 0.3214 > 0.05, indicating no publication bias.

#### 3.6.2. Meta-Analysis of PFS of Immunotherapies

Data on progression-free survival were available for seven studies in the immunotherapy group. Individual PFS and their confidence intervals were plotted for each study within the two groups (Figure 13). The median of PFS was 8.56 months (range: 2.5–12.3 months) (95% CI: 6.40–10.73) in the immunotherapy group.

The meta-analysis yielded a pooled PFS of 6.20 for the targeted drug treatment and 6.33 for the targeted combination chemotherapy group. Still, the *p*-value for the subgroup analysis was 0.60, which was not statistically different, with a considerable heterogeneity of *I*^2^ = 68% (Figure 14).

The funnel plot is asymmetric with more results falling on the left side, suggesting possible publication bias (Figure 15).

The pooled PFS for obtaining immune combination chemotherapy drug treatment was 9.89 months, *p*-value = 0.6 for subgroup analysis, concluding that there was no significant difference in PFS between the two groups (Figure 16).

The funnel plot is asymmetric, with more results falling on the right side, suggesting possible publication bias (Figure 17).

### 3.7. Meta-Analysis of ORR

The meta-analysis heterogeneity of chemotherapy drugs combined with immune drugs versus chemotherapy drugs alone was 0%, yielding an OR of 1.622 (Figure 18). So, the combination of chemotherapy immune drugs yielded a higher objective response rate than chemotherapy alone.

The pooled objective response rate for the targeted therapy group was 32.1%, *I*^2^ = 78%, *p*-value < 0.01 (Figure 19). The ORR values of the three trials in immunotherapy, which had immunotherapy drugs in combination with chemotherapy drugs to compare with chemotherapy drugs alone, found no significant difference.

### 3.8. Meta-Analysis of DCR

The meta-analysis yielded a pooled DCR of 76.6% (*I*^2^ = 29%) for the immune-combination chemotherapy regimen. In contrast, the pooled disease control rate for targeted agents was 79.1% (see Figure 20 and Figure 21).

### 3.9. Treatment-Related Adverse Events

#### 3.9.1. TRAE of Targeted Therapies

The results of treatment-related adverse events are summarised in Table 7. The top ten of these were neutropenia, thrombocytopenia, anemia, fatigue, diarrhea, leukopenia, neuropathy, rash, hypertension, and hand–foot skin reactions. In the erlotinib group, TRAE was less in the group targeted therapy combined chemotherapy than in the control group with chemotherapy agents alone, especially hand–foot syndrome, which occurred in up to 20 cases in grade 3/4 but not in the combination group. A total of 27 cases of rash occurred in the group with cetuximab and none in the group with chemotherapy alone. Moreover, with cetuximab, the incidence of TRAE (*n* = 72) was greater than in the placebo group (*n* = 33). TRAE incidence was higher in both Ramucirumab and Merestinib compared to the placebo group. TRAE was also higher with panitumumab (*n* = 82) compared to chemotherapy alone (*n* =33), where skin toxicity was 36 cases. In the trial of panitumumab, the combination was higher than chemotherapy alone. In the trial of Moehler et al. [23], there were 18 cases of grade 3/4 TRAE in the sorafenib group, while there were 30 cases in the chemotherapy alone group, which is more than in the combination group.

#### 3.9.2. TRAE of Immunotherapies

Grade 3–4 adverse reactions to immunotherapies are summarised in Table 8 below. Sun (2018) [44] showed no significant difference in TRAEs between the monotherapy and combination groups. the TOPAZ-1 trial by D. Y. Oh [50] concluded that the incidence of grade 3/4 adverse reactions was lower in the durvalumab group than in the placebo group. The incidence of grade 3/4 treatment-related adverse events (TRAEs) was 62.7% in the durvalumab-treated group and 64.9% in the placebo-treated group. The rates of TRAEs leading to treatment discontinuation were 8.9% and 11.4%, respectively.

#### 3.9.3. TRAE of Combined Therapies

In the study by J. Zhou et al. [52] 43% of patients had a TRAE of grade 3 or higher. In the trial of Q. Zhang et al. [53] 84.2% of patients had one TRAE. Fatigue (n = 14), anorexia (n = 8), increased alanine aminotransferase (ALT) (n = 7) or aspartate aminotransferase (AST) (n = 7), rash (n = 6), hypertension (n = 5), and hoarseness (n = 5), were the most common TRAEs of any grade. An amount of 34.2% of patients had grade 3 TRAEs, the most prevalent of which were fatigue (n = 5) and hypertension (n = 3). One patient experienced a grade 4 cerebral hemorrhage as a result of hypertension, while five (13.9%) and one (2.8%) patient experienced dose reductions and treatment suspensions as a result of TRAEs. Due to Lenvatinib-related adverse effects, the dose of four individuals was reduced from 8 mg to 4 mg per day. Due to treatment-related cerebral bleeding, one patient terminated Lenvatinib plus PD-1 inhibitor therapy. There were six postoperative problems among patients who had resection, including two cases of biliary leakage, two cases of pleural effusion, one case of delayed liver function recovery, and one incidence of upper gastrointestinal haemorrhage (Table 9).

## 4. Discussion

This study provides an up-to-date, evidence-based systematic review of all clinical trials published between 2010 and 2022 that include all types of BTC. A rigorous quality assessment and a detailed description of trial design, inclusion criteria, characteristics, control arms, and outcomes served as the foundation for this work. Out of the total 709 studies identified, we examined 32 phase 2 studies and one phase 3 study.

From 2010 to 2020, 18 of the 33 studies of targeted or immunologic agents as first-line agents in patients with BTC had positive results. According to the meta-analyses, the effect of immunotherapy in combination with chemotherapy on patients’ ORR was statistically significant.

None of the four trials of sorafenib (Khoueiry (2012) [24], Khoueiry (2014) [25], Lee (2012) [33], and Moehler (2014) [23]) showed evidence of efficacy of sorafenib as a first-line agent in combination with chemotherapy. Lowery (2019) [37] showed that binimetinib could be safely combined with gemcitabine and cisplatin in advanced BTC. However, the observed efficacy signal was modest and not superior to using gemcitabine plus cisplatin alone. The combination of panitumumab with chemotherapy did not improve ORR, PFS, or OS in patients with advanced BTC with KRAS WT. According to a meta-analysis by Vogel (2018) [40], EGFR receptor antagonists did not reveal a benefit on sustained patient survival compared to chemotherapy alone only benefit. Therefore, no further studies investigating the combination of chemotherapy with anti-EGFR antibodies are needed. In contrast, Lau (2018) [28] showed that Everolimus showed clinical activity as first-line monotherapy for advanced BTC in unselected patients.

In three phase 2 trials, sorafenib did not improve survival in this setting. Similarly, other EGFRs, such as Vandetanib and Cetuximab, also failed to improve the prognosis of patients. All these trials suggest that adding EGFR-targeting agents to GEMOX is feasible and safe but ineffective. The lack of efficacy may be related to the heterogeneity of the target population for advanced BTC, the suboptimal treatments explored, or the need for alternative endpoints after survival. 

From the results, it appears that Malka (2014) [29], Lee (2013) [32], Valle (2020) [20], and Leone (2016) [36] all showed that targeted agents were able to improve PFS compared to chemotherapy alone. Still, only one agent, Merestinib, showed a longer OS, while Oh (2020) [50] achieved median OS of 18.1 and 20.7 months among immunotherapies.

Two studies on the combination of immunotherapy and targeted therapy were included. Preliminary data showed that lenvatinib in combination with PD-1 inhibitors showed some efficacy in patients with advanced ICC. Both pembrolizumab and nivolumab showed antitumour effects when combined with lenvatinib. The effect of this combined regimen on overall survival in individuals with advanced ICC is still being studied in clinical trials.

Despite the low response rate of targeted and immunotherapy in BTC and the scarcity of clinical trial data, more research is needed, and better individualised therapy as well as drug combinations may be the way forward for such promising antitumour agents.

It needs to indicate that, in the included studies, outcome data were not counted separately according to the patient’s site of tumour development. They therefore could not be compared based on differences in the anatomical characteristics of the biliary tract.

BTC has an immunogenic profile, implying that immunotherapy is promising. However, current studies show that immune checkpoint inhibitors have limited activity in first-line therapy. With three drugs approved for marketing as second-line therapy, targeted agents have shown some success, but evidence of efficacy as first-line therapy is still lacking. More high-quality RCTs based on patient target genotyping are required to investigate the efficacy of using targeted agents as first-line BTC therapy. This article summarises and analyses current clinical trials in which immune or targeted agents have been added to standard BTC treatment as first-line therapy. Most clinical trials for targeted or immune agents as first-line treatments for BTC are still in the early stages, and future results will provide more evidence for future research. Future multi-institutional clinical trials should allow for large-scale studies that stratify patients based on anatomical subtype and genetic drivers to predict response and prognosis to new treatment regimens.

The drawback of this study is that our conclusions are based on some unadjusted analyses and may be influenced by additional confounding factors, including gender, age, genotypic mutation status, prior systemic medication, and other characteristics. Second, we were unable to conduct further subgroup analyses to assess the efficacy and safety of immunotherapy or targeted therapies in patients with various conditions due to a lack of etiological data.

## 5. Conclusions

This study demonstrates that immunotherapy combined with chemotherapy as first-line treatment can provide survival benefits by improving the objective response rate for patients with unresectable BTC. The potential for combination therapy to become a new trend in clinical treatment is promising. However, because the research design of existing clinical trials is insufficient, more comparable and high-quality investigations of regimens based on immunotherapies or targeted therapy as first-line treatment are required.

## Figures and Tables

**Figure 1 cancers-15-00039-f001:**
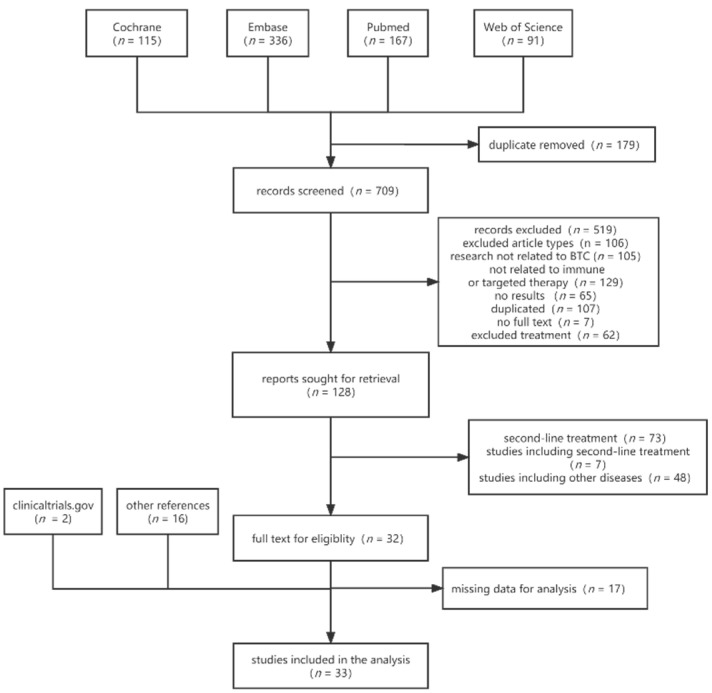
Flowchart of study selection.

**Figure 2 cancers-15-00039-f002:**
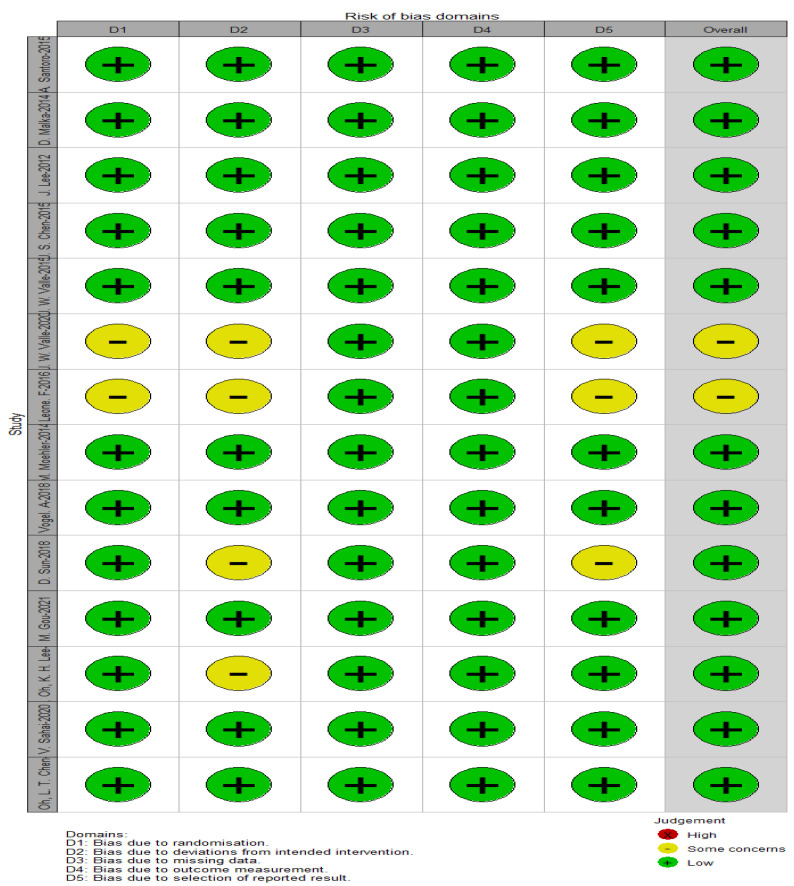
Assessment of the risk of bias using ROB-2.

**Figure 3 cancers-15-00039-f003:**
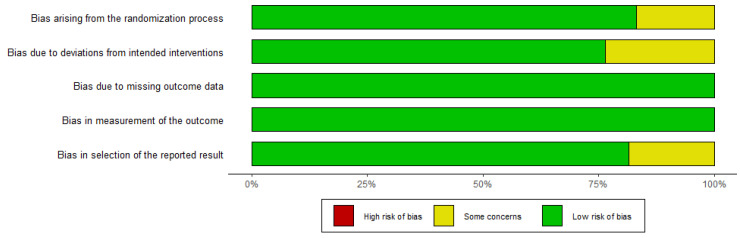
Assessment of the risk of bias using ROB-2 (traffic light figure).

**Figure 4 cancers-15-00039-f004:**
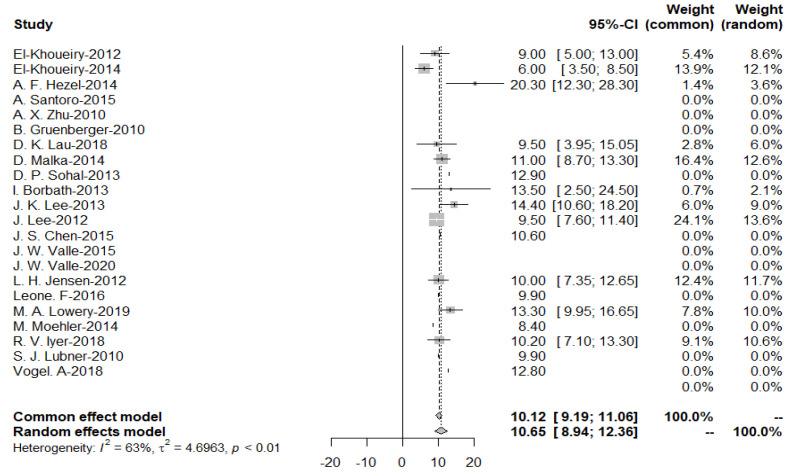
Pooled overall survival with targeted therapy.

**Figure 5 cancers-15-00039-f005:**
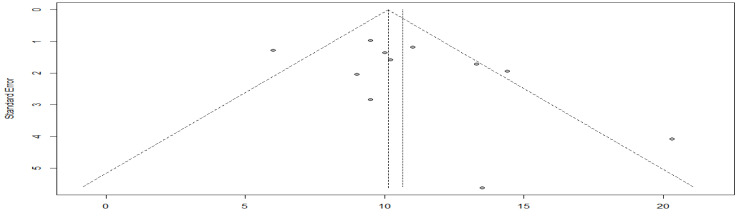
Funnel plot of overall survival with targeted therapy.

**Figure 6 cancers-15-00039-f006:**
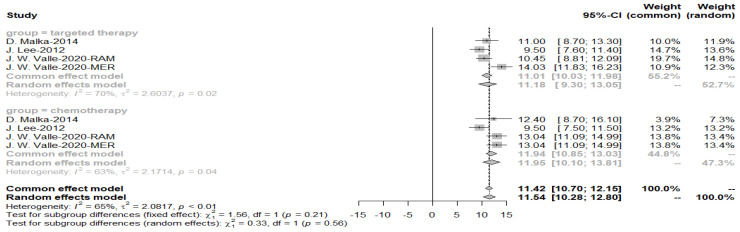
Overall survival in the targeted chemotherapy combination group compared with the chemotherapy alone group.

**Figure 7 cancers-15-00039-f007:**
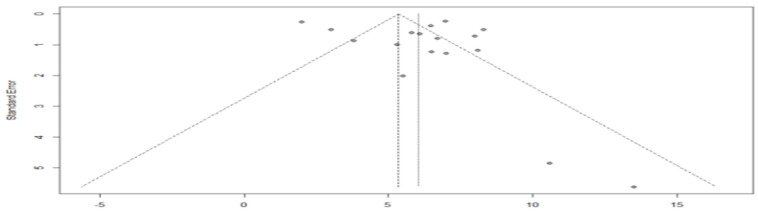
Funnel plot of overall survival in the targeted chemotherapy combination group compared with the chemotherapy agent alone group.

**Figure 8 cancers-15-00039-f008:**
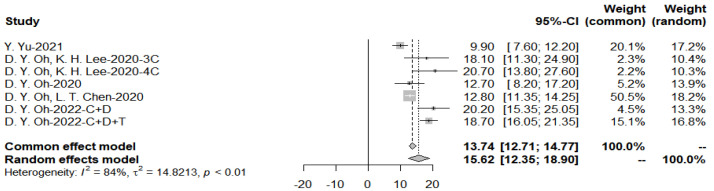
Pooled overall survival with immunotherapy combined with chemotherapy.

**Figure 9 cancers-15-00039-f009:**
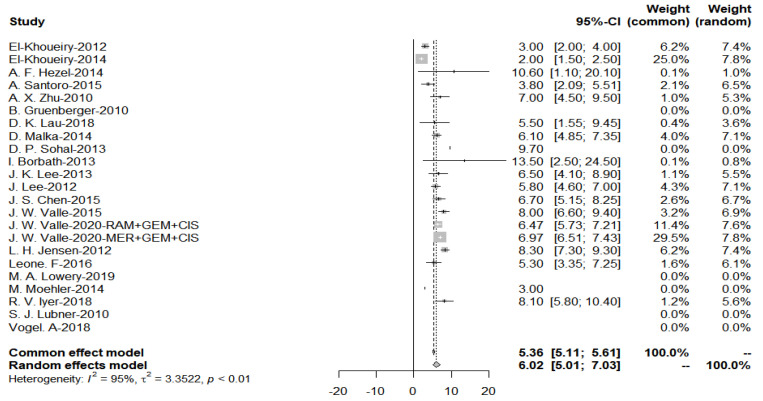
Progression-free survival with targeted therapy.

**Figure 10 cancers-15-00039-f010:**
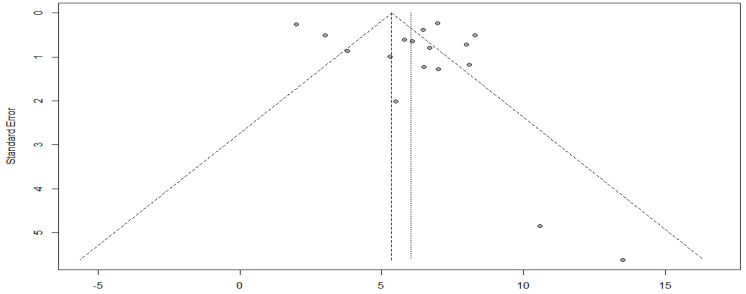
Funnel plot of progression-free survival with targeted therapy.

**Figure 11 cancers-15-00039-f011:**
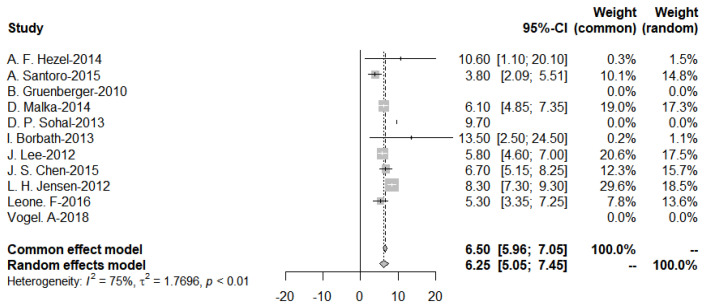
Progression-Free-Survival of EGFR targeted therapies.

**Figure 12 cancers-15-00039-f012:**
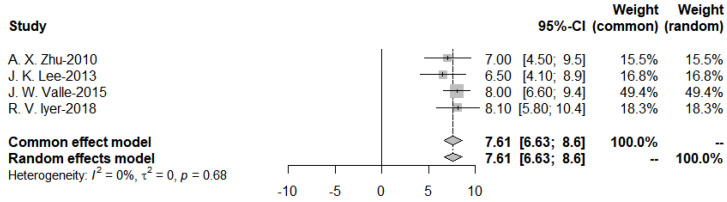
Progression-Free-Survival of VEGF targeted therapies.

**Figure 13 cancers-15-00039-f013:**
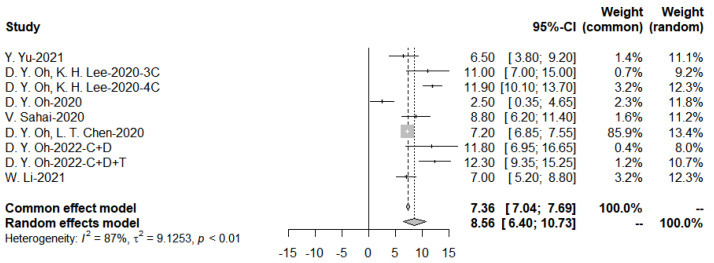
Progression-free survival with immunotherapy and funnel plot.

**Figure 14 cancers-15-00039-f014:**
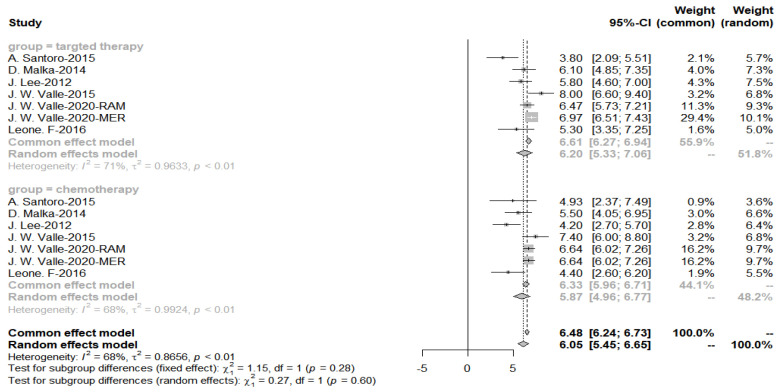
Progression-free survival in the targeted chemotherapy combination group compared with the chemotherapy alone group.

**Figure 15 cancers-15-00039-f015:**
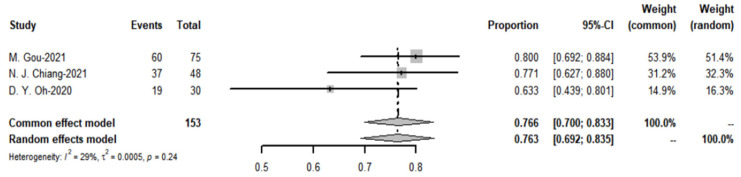
Funnel plot of progression-free survival in the targeted chemotherapy combination group compared with the chemotherapy alone group.

**Figure 16 cancers-15-00039-f016:**
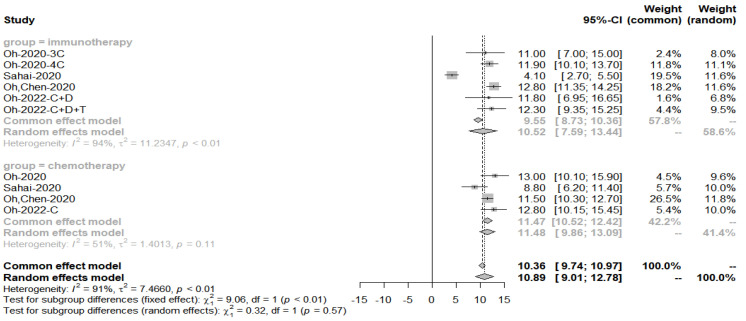
Progression-free survival in the immuno-chemotherapy combination group compared to the immunotherapy group.

**Figure 17 cancers-15-00039-f017:**
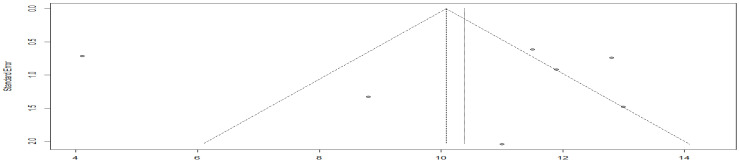
Funnel plot of progression-free survival in the immuno-chemotherapy combination group compared to the immunotherapy group.

**Figure 18 cancers-15-00039-f018:**
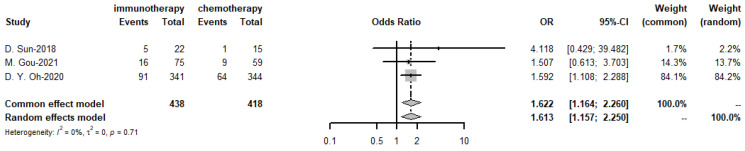
Objective response rate in the immuno-chemotherapy combination group compared to the chemotherapy drug alone group.

**Figure 19 cancers-15-00039-f019:**
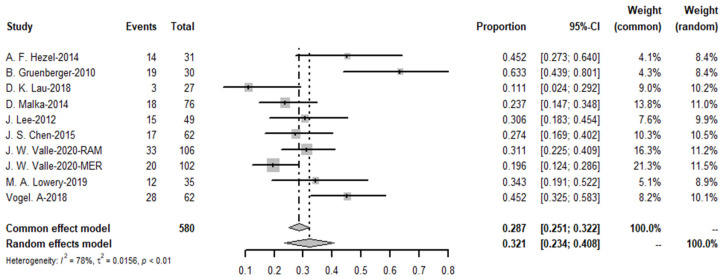
Objective response rate of targeted therapies.

**Figure 20 cancers-15-00039-f020:**
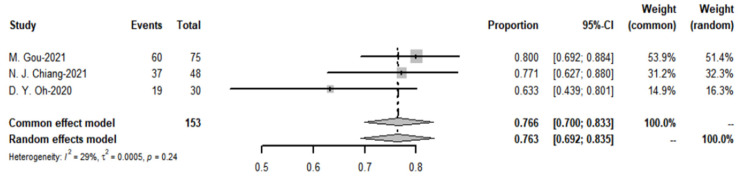
Disease control rate of targeted therapies and funnel plot.

**Figure 21 cancers-15-00039-f021:**
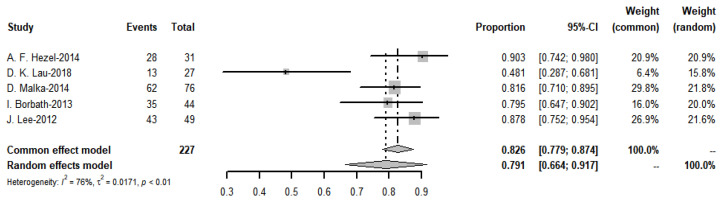
Disease control rate of immunotherapies and funnel plot.

**Table 1 cancers-15-00039-t001:** Types of interventions.

Targets	Drugs
MEK1/2	Binimetinib
EGFR	Panitumumab, Cetuximab, Vandetanib, Erlotinib
VEGF	Bevacizumab, Cediranib, Ramucirumab
mTOR	Everolimus
PD-1/PD-L1	Pembrolizumab, Nivolumab, Bintrafusp, Camrelizumab, Durvalumab
Multi targets	Sorafenib, Bintrafusp Alfa, Lenvatinib, Toripalimab
MET	Merestinib
CTLA-4	Tremelimumab, Ipilimumab

**Table 3 cancers-15-00039-t003:** Primary target sites of targeted-therapy studies.

Disease Type	Patients
IHC	694
EHC	235
Cholangiocarcinoma	219
Gallbladder Cancer	384
Perihillar	44
Vater ampulla carcinoma	17
Other	48

**Table 4 cancers-15-00039-t004:** Basic characteristics of immunotherapies.

No.	Study	Phase	Patients	Intervention	Dose	Chemotherapy Type	Age	Gender
1	Yu (2021) [43]	II	14	Camrelizumab	Camrelizumab 3 mg/kg d1, Q2 W or Q3 W	different chemotherapy regimens	median age: 50.5 (36–70)	male: 71.43%
2	Sun (2018) [44]	II	−15 −22 (C + I)	PD-1 inhibitors	1 cycle of Gem 1000 mg/m^2^ + Cis 25 mg/m^2^ on D1 and D8, followed by GEMCIS + D 1120 mg and T 75 mg, Q3W	Gemcitabine-based (*n* = 12) Paclitaxel-Albumin-based (*n* = 4) Oxaliplatin + tegafur (n = 2) Other (*n* = 1)	average age > 65	NA
3	Gou (2021) [45]	II	−59 −75 (C + I)	PD-1 inhibitors (Pembrolizumab, Nivolumab, Sintilimab, Toripalimab)	SHR-1210 3mg/kg and Gemcitabine 800 mg/m^2^ will be administered IV Q2W	different chemotherapy regimens	NA	male: 67.2%
4	Oh (2020) [46]	II	121	Durvalumab (D) ± Tremelimumab (T)	Durvalumab (1500 mg every 3 weeks [Q3W]) or placebo + GEMCIS (Gem 1000 mg/m^2^ and Cis 25 mg/m^2^	GEMCIS	NA	NA
5	Chen (2020) [47]	II	38	Camrelizumab		GEMOX	NA	NA
6	Chiang (2021) [48]	II	48	Nivolumab		Gemcitabine and S-1	NA	NA
7	Oh (2020) [41]	II	30	Bintrafusp alfa	Bintrafusp alfa 1200 mg every 2 weeks	GEMCIS	NA	NA
8	Sahai (2020) [49]	II	71	Ipilimumab Nivolumab		GEMCIS	median age: 62 (20–80)	male: 49%
9	Oh (2020) [50]	III	−341 −344 (placebo)	Durvalumab	durvalumab (1500 mg every 3 weeks [Q3W]) or placebo + GEMCIS (Gem 1000 mg/m^2^ and Cis 25 mg/m^2^ on Days 1 and 8 Q3W) for up to 8 cycles	GEMCIS	64	male: 50.4%
10	Oh (2022) [42]	II	−30 (chemo) −47 (C + D) −47 (C + D) 47 (C + D + T)	Durvalumab Tremelimumab	Gemcitabine 1000 mg/m^2^ plus Cisplatin 25 mg/m^2^ Durvalumab 1120 mg Tremelimumab 75 mg	GEMCIS	median age: 64 years (58–70)	male: 49%
11	Li (2021) [51]	II	15	Toripalimab	Toripalimab (240 mg intravenously every three weeks) gemcitabine 1000 mg/m^2^ d1, d8 + S-1 40–60 mg bid D1-14, Q21d	Gemcitabine	median age: 62 years	male: 56%

**Table 5 cancers-15-00039-t005:** Primary target sites of immunotherapy studies.

Disease Type	1 [41]	2 [42]	3 [43]	10 [40]	Summary
Gallbladder	2	8	NA	171	181
Cholangiocarcinoma	NA	69	NA	NA	69
Intrahepatic	9	NA	74	377	460
Extrahepatic	3	NA	60	130	193

**Table 6 cancers-15-00039-t006:** ROBINS-1 of non-RCT (risk of bias).

Study	R1 *	R2	R3	R4	R5	R6	R7
Khoueiry (2012) [24]	Moderate	Moderate	Moderate	Low	High	High	Low
Khoueiry (2014) [25]	Low	Moderate	Low	Moderate	High	Moderate	Low
Hezel (2014) [26]	Moderate	Low	Low	Low	Low	Low	Moderate
Zhu (2010) [52]	Low	Low	Low	Low	Low	Low	Low
Gruenberger (2010) [27]	Low	Low	Low	Low	Low	Low	Low
Lau (2018) [28]	Moderate	Low	Low	Low	Moderate	Low	Low
Sohal (2013) [30]	Low	Low	Low	Low	Low	Low	Low
Borbath (2013) [31]	Low	Low	Low	Low	Low	Low	Low
Lee (2013) [32]	Low	Low	Moderate	Low	Moderate	Low	Low
Lowery (2019) [37]	Low	Low	Low	Low	Low	Low	Low
Lubner (2010) [39]	Low	Low	Low	Low	Low	Low	Low
Yu (2021) [43]	Moderate	Low	Low	Low	Low	Low	Low
Oh (2020) [50]	Moderate	Moderate	Moderate	Moderate	Moderate	Moderate	Moderate
Chen (2020) [47]	Low	Low	Low	Low	Low	Low	Low
Chiang (2021) [48]	Low	Low	Low	Low	Low	Low	Low
Jensen (2012) [35]	Low	Low	Low	Low	Low	Low	Low
Moehler (2014) [23]	Low	Low	Low	Low	Low	Moderate	Low

* R1: Confounding R2: Selection bias R3: Bias in measurement classification of interventions R4: Bias due to deviations from intended interventions R5: Bias due to missing data R6: Bias in measurement of outcomes R7: Bias in selection of the reported result.

**Table 7 cancers-15-00039-t007:** Treatment-Related Adverse Events of targeted therapies.

TRAE	Arm A	Arm B			
**Malka-2014**	Cetuximab + GEMOX	GEMOX	**J. W. Valle-2015**	Cediranib	placebo
peripheral neuropathy	18	10	hypertension	23	13
neutropenia	17	11	diarrhoea	8	2
increased aminotransferase	17	10	platelet count decreased	10	4
**J. Lee-2012**	GEMOX + Erlotinib	GEMOX	white blood cell decreased	15	7
Nausea	1	3	fatigue	16	7
Vomiting	0	4	neutropenia	52	33
Diarrhoea	5	1	thrombocytopenia	37	17
Stomatitis	1	0	anemia	29	19
Constipation	0	0	**J. W. Valle-2020**	Merestinib	placebo
hand-foot syndrome	0	20	neutropenia	48	33
Neutropenia	3	5	thrombocytopenia	17	17
Thrombocytopenia	3	0	alanine aminotransferase (ALT) increased	11	5
Raised AST	3	4	**J. W. Valle-2020**	Ramucirumab	placebo
Raised ALT	3	4	neutropenia	52	33
Skin rash	3	0	thrombocytopenia	37	17
Neuropathy	1	0	anemia	29	5
Asthenia	1	2	Leone. F-2016	GEMOX + Panitumumab	GEMOX
Anorexia	3	1	skin toxicity	36	6
Mucositis	0	0	diarrhea	25	14
Pruritus	0	0	mucositis	10	6
**J. S. Chen-2015**	C-GEMOX	GEMOX	Constipation	11	7
Neutropenia	11	2	**M. Moehler-2014**	gemcitabine + sorafenib	placebo
Thrombocytopenia	8	2	Fatigue	1	2
Oral mucositis	2	1	Thrombocytopenia	4	6
Diarrhea	2	2	Hand-foot syndrome	0	7
Nausea	0	2	Diarrhea	0	1
Vomiting	0	0	Leukopenia	2	4
Fatigue	2	2	Rash	0	0
ALT increased	2	1	Oral disorder	0	0
Anorexia	2	5	Nausea	4	2
Neuropathy	5	5	Alopecia	0	0
Allergic reaction	1	0	Anaemia	1	2
Skin rash	27	0	Stomatitis	0	0
**Vogel. A-2018**	GEMCIS + panitumumab	GEMCIS	Vomiting	1	2
Leucopenia	8	13	Pruritus	0	0
Neutropenia	13	26	Epistaxis	0	0
Febrile neutropenia	0	3	Fever	1	2
Thrombopenia	12	18	Neutropenia	4	2
Anemia	3	7	Obstipation	0	0
Dry Skin	0	3			
Nail changes	0	1			
Rash	0	7			
Acne	0	10			
Diarrhea	0	3			
Mucositis	1	0			
Nausea	1	2			
Fatigue	0	4			
Fever	0	0			
Infection	6	6			
Neuropathy	0	0			
Dyspnea	0	1			
			**total**	669	476

**Table 8 cancers-15-00039-t008:** Treatment-related adverse events of immunotherapies.

Y. Yu-2021	Arm A	Arm B
Vomiting	4	
Fever	1	
Anorexia	1	
Drug-induced allergy	1	
Hepatitis	1	
White blood cell count decreased	1	
Aspartic aminotransferase increased	1	
Platelet count decreased	1	
Neutrophil count decreased	1	
**D. Sun-2018**	combination group	monotherapy group
thrombocytopenia	5	2
leukopenia	3	
**M. Gou-2021**		
hypothyroidism	3	
rash	2	
hepatitis	1	
leukopenia	3	
**D. Y. Oh-2020**		
neutropenia	66	
nausea	72	
pruritus	67	
anemia	43	
thrombocytopenia	20	
**X. Chen-2020**		
fatigue	27	
fever	27	
hypokalemia	7	
**N. J. Chiang-2021**	nivolumab and gemcitabine and S-1	gemcitabine and S-1
skin toxicity	17	7
hypothyroidism		3
hypophysis		3
pneumonitis		3
**D. Y. Oh, F. de Braud-2020**		
rash		
maculopapular rash	5	
fever	4	
increased lipase	3	

**Table 9 cancers-15-00039-t009:** Treatment-related adverse event of combined therapies.

TRAE, n	All Grades	Grade 1	Grade 2	Grade 3	Grade 4
All	32	28	19	12	1
Fatigue	14	7	2	5	NA
Anorexia	8	8	NA	NA	NA
ALT elevation	7	7	NA	NA	NA
AST elevation	7	6	1	NA	NA
Rash	6	NA	4	2	NA
Hypertension	5	1	NA	3	1
Hoarseness	5	5	NA	NA	NA
Leukopenia	4	2	2	NA	NA
Erythrocytopenia	4	4	NA	NA	NA
Muscle soreness	4	1	3	NA	NA
Pruritus	4	NA	2	2	NA
Hand and foot syndrome	4	1	3	1	NA
Anemia	3	3	NA	NA	NA
Nausea	3	NA	3	NA	NA
Fever	3	1	2	NA	NA
Diarrhea	3	2	1	NA	NA
Hypothyroidism	3	NA	3	NA	NA
Alkaline phosphatase increased	3	3	NA	NA	NA
Weight loss	3	3	NA	NA	NA
Alopecia	3	3	NA	NA	NA

## Data Availability

The data presented in this study are available on request from the corresponding author.

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
