# Peer review of "Efficacy and Safety of First-Line Targeted Treatment and Immunotherapy for Patients with Biliary Tract Cancer: A Systematic Review and Meta-Analysis"

_cancers, 2022, doi:10.3390/cancers15010039_

Round 1

Reviewer 1 Report

Comments to the authors:

Major comments

1.      I would add a paragraph in which briefly discuss about the molecular targets and explain the concept of agnostic therapies.

2.      The English is extremely poor in terms of grammar, punctuation, and syntax (i.e., randomised instead of randomized, data were collected instead of data was collected, chemotherapies instead of chemotherapy, leaving 128 articles instead of identifying 128 articles, and more). The paper is difficult to read and understand.

3.      A definition of objective response rate (ORR), disease control rate (DCR), overall survival (OS), and progression-free survival (PFS is necessary. Also, how these are calculated should be clarified.

Minor comments

4.      This sentence “Only EGFR inhibitors were studied in Alexandro Rizzo's study. Although data on treatment, including systemic chemotherapy and radiation therapy, were analyzed in Michael N's study, the data were only available up to 2013 and did not focus on targeted therapies. As a result, there is a need for more recent and comprehensive meta-analysis in this field”, at page 2, needs references.

5.      When describing the type of studies included, I would say: “we focused on RCT and cohort studies.

6.      I would not use the word “yellow” when talking about race.

7.      Please, review the use of capital letters and the format of references in the tables. The layout of the tables 3 and 5 is very confusing and should be improved. Table 2 is shown after table 6.

8.      The first sentence at page 7 is confusing.

9.      Abbreviations are used inconsistently (i.e., table 6, targeted therapies at page 10).

10.  When citing a paper, it is not sufficient to report the name of the first author, it would be better to say, “the trial by… et al” or “… et al. showed that…”.

11.  The first sentence at page 11 is impossible to understand.

12.  The numeration of figures is completely wrong. 

Author Response

Responses to Reviewer 1 Comments

Thanks very much for taking the time to review this manuscript. We sincerely appreciate all your comments and suggestions! Please find our itemised responses below and revisions/corrections in the re-submitted manuscript.

Major comments

  1. I would add a paragraph in which briefly discuss about the molecular targets and explain the concept of agnostic therapies.

Response 1: Thank you for your advice! We have added more information in the Introduction section to explain the molecular targets of biliary tract cancer. Please see Page 2 Line 55-75.

  1. The English is extremely poor in terms of grammar, punctuation, and syntax (i.e., randomised instead of randomised, data were collected instead of data was collected, chemotherapies instead of chemotherapy, leaving 128 articles instead of identifying 128 articles, and more). The paper is difficult to read and understand.

Response 2: We apologise for the poor language of our manuscript. Based on these comments and suggestions, we have tried our best to work on both language and readability and involved native English speakers in language corrections. In addition, we have carefully modified and proofread the original manuscript to minimise typographical and grammatical errors. We hope you will agree with our language improvement.

  1. A definition of objective response rate (ORR), disease control rate (DCR), overall survival (OS), and progression-free survival (PFS is necessary. Also, how these are calculated should be clarified.

Response 3: Following your comment, more information about these concepts has been included in the revised manuscript. Please see Page 4, Line 133-143.

Minor comments

  1. This sentence “Only EGFR inhibitors were studied in Alexandro Rizzo's study. Although data on treatment, including systemic chemotherapy and radiation therapy, were analysed in Michael N's study, the data were only available up to 2013 and did not focus on targeted therapies. As a result, there is a need for more recent and comprehensive meta-analysis in this field”, at page 2, needs references.

Response 4: Thanks for your suggestion! We have added the references.

  1. When describing the type of studies included, I would say: “we focused on RCT and cohort studies.

Response 5: We have revised the text according to your suggestion.

  1. I would not use the word “yellow” when talking about race.

Response 6: We were sorry for the improper wording. It has been modified to “Asians”.

  1. Please, review the use of capital letters and the format of references in the tables. The layout of the tables 3 and 5 is very confusing and should be improved. Table 2 is shown after table 6.

Response 7: Based on your comment, we have made the corrections to make the unit harmonised within the whole manuscript.

  1. The first sentence at page 7 is confusing.

Response 8: According to your suggestion, we have modified “The regimens included in the study were all immunotherapy drugs combined with chemotherapy drugs as first-line treatment, and no immunotherapy drugs were used alone.” into “No immunotherapy drugs were used alone in any of the study's regimens, which all included both immunotherapy and chemotherapy as the first line of treatment.”

  1. Abbreviations are used inconsistently (i.e., table 6, targeted therapies at page 10).

Response 9: Abbreviations used in each table have been modified and defined in a note at the bottom of the tables.

  1. When citing a paper, it is not sufficient to report the name of the first author, it would be better to say, “the trial by… et al” or “… et al. showed that…”.

Response 10: We have corrected these mistakes based on your suggestions.

  1. The first sentence at page 11 is impossible to understand.

Response 11: It has been modified to “This study concluded that the combination of bevacizumab and erlotinib demonstrated significant activity in the treatment of advanced biliary tract cancer, with few adverse events of grades 3 or 4.”

  1. The numeration of figures is completely wrong.

Response 12: We changed it to the correct order. We sincerely apologise for this careless mistake.

Thanks again for your professional help!

Reviewer 2 Report

The authors evaluated targeted treatment and immunotherapy for patients with BTC by systematic review and meta-analysis, and suggested that combination therapy could become a new trend in treatment for advanced BTC. This is a backbreaking study and could be a promising approach to unresectable BTC patients.

Comments

1. Recent WHO tumor classification of the biliary tract reported that BTC showed characteristic clinicopathological features and genetic and molecular changes along the biliary tract. While authors cited this tumor classification briefly in the introduction  section, there are no discussion of targeted treatment and immunotherapy in BTCs according to the biliary anatomy. Please discuss on this point citing references, if possible, using the data of this review and meta-analytic data.

2. In the section of 3.4. systemic review, the authors analyzed targeted treatment and immunotherapy separately, though there are no systemic review of combined treatment. Please describe 3.4.3 for combined treatment. 

In the section of 3.10. treatment-related adverse events, there is no TRAE for combined treatment. Please describe 3.10.3. for combined treatment.

3. In page 3. Unresectable or metastatic gallbladder ---.  Metastatic is confusing. Only "unresectable" is enough.

4. In page 5. First two lines are difficult to understand.

5. In page 6, what is jugular carcinoma ?

6. Are there superiority of combined treatment to either of targeted treatment or immunotherapy ? Please discuss the superiority in the Discussion section.

Author Response

Response to Reviewer 2 Comments

Thanks very much for taking the time to review this manuscript. We sincerely appreciate all your comments and suggestions! Please find our responses below and revisions/corrections in the re-submitted manuscript.

  1. Recent WHO tumor classification of the biliary tract reported that BTC showed characteristic clinicopathological features and genetic and molecular changes along the biliary tract. While authors cited this tumor classification briefly in the introduction section, there are no discussion of targeted treatment and immunotherapy in BTCs according to the biliary anatomy. Please discuss on this point citing references, if possible, using the data of this review and meta-analytic data.

Response 1: Thank you for your suggestion! In the studies we included, there was essentially no separate data on outcome indicators by type of primary site. We have added this information to the Discussion section. Please see Page 24 Line 566-568.

  1. In the section of 3.4. systemic review, the authors analysed targeted treatment and immunotherapy separately, though there are no systemic review of combined treatment. Please describe 3.4.3 for combined treatment.

In the section of 3.10. treatment-related adverse events, there is no TRAE for combined treatment. Please describe 3.10.3. for combined treatment.

Response 2: Thank you for underlining this deficiency! Following your comment, we have added 3.4.3 and 3.9.3 to the revised manuscript.

  1. In page 3. Unresectable or metastatic gallbladder ---.  Metastatic is confusing. Only "unresectable" is enough.

Response 3: This sentence was rephrased according to the comment.

  1. In page 5. First two lines are difficult to understand.

Response 4: We have modified the sentence according to the comment. It has been changed to “This study included 22 trials focusing on targeted therapies, with a total of 1,658 patients.”

  1. In page 6, what is jugular carcinoma ?

Response 5: It should be vater ampulla carcinoma. We have corrected the mistake.

  1. Are there superiority of combined treatment to either of targeted treatment or immunotherapy ? Please discuss the superiority in the Discussion section.

Response 6: Thank you for the suggestion! Unfortunately, this conclusion could not be made, due to the small sample size. The two available studies suggest that combination therapy is promising but has a high incidence of adverse events. We have added the information in the discussion. Please see Page 24 Line 557-562.

Thanks again for your professional help!

Round 2

Reviewer 1 Report

The revised version improve significantly the manuscript

Thanks for the opportunity to review this article